# Survey of Colistin Resistance in Commensal Bacteria from *Penaeus vannamei* Farms in China

**DOI:** 10.3390/foods12112143

**Published:** 2023-05-26

**Authors:** Yilin Zhang, Xinrui Lv, Weiwei Cao, Huang Zhang, Lei Shi, Weibin Bai, Lei Ye

**Affiliations:** 1Institute of Food Safety and Nutrition, Jinan University, Guangzhou 510632, China; stzhyl@163.com (Y.Z.); lxrui_1995@163.com (X.L.); shilei@jnu.edu.cn (L.S.); baiweibin@163.com (W.B.); 2School of Food Science and Biology, Guangdong Polytechnic of Science and Trade, Guangzhou 510640, China; weiwei09201029@163.com; 3Guangzhou Double Helix Gene Technology Co., Ltd., Guangzhou International Bio Island Co., Ltd., Guangzhou 510320, China; 18680288520@163.com; 4Shandong Yuwang Ecological Food Industry Co., Ltd., Yucheng 251200, China

**Keywords:** antibiotic resistance, aquatic shrimp, prevalence, whole-genome sequencing, genome-wide association study

## Abstract

Aquatic environments are important reservoirs for drug resistance. Aquatic foods may act as carriers to lead antibiotic-resistant commensal bacteria into the human gastrointestinal system, then contacting gut microbiota and spreading antibiotic resistance. Here, several shrimp farms were investigated to identify colistin resistance among commensal bacteria of aquaculture. A total of 884 (41.6%) colistin-resistant isolates were identified among 2126 strains. Electroporation demonstrated that colistin-resistant fragments were present in some commensal bacteria that could be transferred to other bacteria. Most of the resistant bacteria were *Bacillus* spp., with 69.3% of the *Bacillus* species exhibiting multiple drug resistance. *Bacillus licheniformis* was prevalent, with 58 strains identified that comprised six sequence types (ST) based on multilocus sequence typing. Whole-genome sequencing and comparisons with previous *B. licheniformis* genomes revealed a high degree of genomic similarity among isolates from different regions. Thus, this species is widely distributed, and this study provides new insights into global antibiotic-resistant characteristics of *B. licheniformis*. Sequence analyses further revealed some of these strains are even pathogenic and virulent, suggesting the antibiotic resistance and hazards of commensal bacteria in aquaculture should be considered. Considering the “One Health” perspective, improved monitoring of aquatic food is needed to prevent the spread of drug-resistant commensal bacteria from food-associated bacteria to humans.

## 1. Introduction

Commensal bacteria live with other organisms and are generally benign. Commensal bacteria are abundant and diverse in biosomes and environments, where they play important roles in the functioning of their host systems. Many previous studies on commensal bacteria have focused on health-threatening pathogens. However, a more recent comprehensive understanding of commensal bacteria has revealed that they interact in numerous ways aside from pathogenesis. For example, commensal bacteria exist in human gastrointestinal tracts, oral cavities, and skin, where they can regulate human immune systems [1,2,3]. When immune functioning decreases, commensal bacteria can cause diseases [4]. Notably, commensal bacteria play key roles in the emergence and spread of antibiotic resistance [5]. For example, multidrug-resistant commensal bacteria from wild birds in Florida (i.e., Amazon parrots) were identified to have spread to ducks in eastern China [6]. Indeed, multidrug-resistant commensal bacteria can spread to other bacterial communities in different environments, including the food we eat, thereby potentially spreading difficult-to-treat infections globally [7].

The rapid increase of antibiotic resistance within global ecosystems has become a major public health problem due to increased use of antimicrobials. Indeed, the emergence and considerable spread of antibiotic resistance have compromised the efficacy of some drugs, hindered progress in antimicrobial therapy, and limited treatment options [8]. Polymyxin E (colistin) is designated by the World Health Organization (WHO) as one of the most important antibiotics in human medicine and is mainly used to treat complex infections caused by multidrug-resistant bacteria [9]. Studies on colistin resistance have primarily focused on opportunistic pathogens such as *Enterobacteriaceae* and enterococci, while commensal bacteria are rarely evaluated [10,11,12]. Considering the ubiquity of commensal bacteria, investigating colistin-resistant commensal bacteria can provide useful insights into overall colistin resistance.

Aquatic environments are important hosts of antibiotic-resistant bacteria. Indeed, some commensal bacteria associated with aquatic products and their environments have been observed to exhibit antibiotic resistance [9,13]. Colistin is not officially approved for aquaculture use in China, but aquatic environments may be contaminated with colistin via farm runoff that carries animal manure containing residual colistin [14]. When antibiotics are released into aquatic environments, they can be transported via water or retained in sediments, thereby affecting the behaviours of aquatic organisms [7]. Importantly, colistin resistance can spread between animals and environments, and even to humans through direct contact or through consumption of colistin-contaminated foods [12]. Thus, aquatic food should be incorporated into surveillance programs based on the recommendations of the “One Health” program, which will provide new insights into strategies to combat antimicrobial resistance [15]. China has the highest abundance of *Penaeus vannamei* farms based on the Food and Agriculture Organisation (FAO) of the United Nations [16]. The Chinese shrimp aquaculture industry is a major component of China’s aquaculture and is continuously expanding, indicating that its overall impacts on environments and food safety will also increase.

In this study, the prevalence of colistin resistance was investigated among commensal bacteria isolated from different *Penaeus vannamei* farms from the four primary shrimp farming provinces in China. Characterisation of the colistin-resistant commensal bacteria of aquatic shrimp provides new insights into the spread of antibiotic resistance from aquatic food and the potential risks to public health.

## 2. Materials and Methods

### 2.1. Samples and Bacterial Isolation

The four primary shrimp farming provinces of China were sampled from 2019 to 2022, including the Guangdong, Fujian, Jiangsu, and Shandong areas. Shrimp, soil, aquafeed, and pond water samples were collected from a total of 18 *P. vannamei* farms. The breeding scale of each shrimp pond was approximately 50,000–100,000, and shrimp were all in the grow-out stage. All samples were individually collected into sterile plastic bags and immediately transported at ≤4 °C to the laboratory, followed by analysis within 24 h of sampling.

Samples were diluted in 0.9% (*w*/*v*) NaCl to obtain 10-fold serial dilutions. The diluted supernatants (100 µL) were then spread on CHROMagar Orientation plates supplemented with colistin (2 µg/mL). Colonies with different characteristics were identified, purified, and sub-cultured from each sample.

### 2.2. Colistin Resistance Screening

All strains were subjected to analysis of minimum inhibitory concentrations (MICs) for colistin. The broth microdilution (BMD) method was then used based on the ISO standard 20776-1 [17]. An experimental quality control was evaluated based on comparison to the colistin-susceptible strain *Escherichia coli* ATCC 25922 [14]. Colistin concentrations ranged from 2 to 1024 µg/mL. Isolates with a colistin threshold of >2 µg/mL were considered resistant based on European Committee on Antimicrobial Susceptibility Testing (EUCAST) recommendations [18].

### 2.3. 16S rRNA Gene Analyses

Drug-resistant strains were selected for 16S rRNA gene sequencing based on proportions of samples. Genomic DNA was extracted using a HiPure Bacterial DNA extraction kit (Magen, Guangzhou, China) following the manufacturer’s suggestions. The universal bacterial primers 27 F (AGAGTTTGATCCTGGCTCAG) and 1492R (GGTTACCTTGTTACGACTT) were used in polymerase chain reactions (PCRs), followed by amplicon product sequencing using Sanger sequencing (Sangon Biotech, Guangzhou, China) to identify bacterial types [19].

### 2.4. Electroporation Transformation

We selected 20 *Bacillus* strains, three *Staphylococcus aureus* strains, two *Aeromonas* strains, and three *Escherichia coli* strains from colistin-resistant bacteria. The strains were selected as commensal bacterial representatives for plasmid extraction using Plasmid Midi Kits (QIAGEN, Hilden, Germany), following the manufacturer’s recommendations. Plasmids were introduced to competent *Escherichia coli* DH5α cells by electroporation transformation. The transformants were then spread on nutrient agar (containing 2 µg/mL colistin) and cultured overnight. The MIC values for colistin in the transformants were determined using the BMD method.

### 2.5. Antimicrobial Susceptibility Testing

Antimicrobial susceptibility tests were conducted using Kirby-Bauer tests for 10 antimicrobial agents, namely, ampicillin (10 μg), ceftiofur (30 μg), ceftriaxone (30 μg), cefepime (30 μg), trimethoprim-sulfamethoxazole (19 μg), meropenem (10 μg), imipenem (10 μg), gentamicin (10 μg), tetracycline (30 μg), and ciprofloxacin (5 μg). The BMD method was used to assess fosfomycin resistance. *Staphylococcus aureus* ATCC 29213 was used for quality control. Antimicrobial susceptibility was interpreted by comparison of data to standards from the EUCAST [18].

The multidrug resistance (MDR) index was calculated as the ratio of the number of antibiotics that the strains were resistant towards divided by the total number of evaluated antibiotics. The MDR value is typically equal to or less than 0.2 when antibiotics are used infrequently or at low doses to treat animals. Contrarily, using antibiotics more frequently or at a higher risk of exposure will result in an MDR index score that is greater than 0.2 [20].

### 2.6. Multilocus Sequence Typing (MLST)

The most representative strains were selected for MLST to investigate genetic relationships. PCR primers and amplification conditions for housekeeping genes were identified from PubMLST, as shown in Appendix A. Alleles and sequence types (STs) were assigned based on the MLST database [21].

### 2.7. Whole Genome Sequencing (WGS) and Analysis

Seventeen *Bacillus licheniformis* strains were selected as representatives for WGS. Dominant bacteria were selected based on 16S rRNA gene analyses, with only *B. licheniformis* chosen for genome sequences. Isolates were then separated based on antibiotic resistance patterns, sources, and STs. The isolates comprised six strains from Guangdong, four from Fujian, three from Shandong, and four from Jiangsu. DNA was extracted from the isolates using the DNeasy Blood and Tissue Kit (Qiagen, Hilden, Germany) according to the manufacturer’s instructions, and the resulting genomic DNAs were sequenced on the Illumina NovaSeq platform at Shanghai Personal Biotechnology Co., Ltd. (Shanghai, China). Standard Illumina TruSeq Nano DNA LT library preparations (Illumina TruSeq DNA Sample Preparation Guide) were used to construct genome libraries. The FastQC software program (version.0.11.7) was used to evaluate the quality of sequence reads. Data assembly was conducted after removing adapters and filtering data using the AdapterRemoval program (version.2.2.2) [22] and SOAPec (version.2.03) [23]. Filtered reads were assembled with SPAdes (version. 3.12.0) [24], and A5-miseq (version 20160825) [25] was used to generate scaffolds and contigs.

The CARD (version 3.2.4) was used to determine resistance genes and known chromosomal mutations conferring antibiotic resistance [26]. Virulence factors were predicted by comparison against the VFDB database [27]. The PathogenFinder program (version 1.1) was also used to predict the pathogenicity of isolates [28]. Insertion sequences (ISs) were predicted by comparison against the ISFinder database, and the ICEfinder database was used to identify transposons and integrative conjugative elements (ICEs) [29,30].

To assess whether any of the isolates from this study were related to previously identified *B. licheniformis*, the genome sequences of available strains and their corresponding metadata were retrieved from GenBank. A total of 73 records from different sources and countries were retrieved that comprised the ST groups ST1 (*n* = 16), ST3 (*n* = 56), and ST4 (*n* = 1).

### 2.8. Phylogenetic Analysis

A *B. licheniformis* phylogeny was constructed using the CSI Phylogeny-1.4 program (https://cge.cbs.dtu.dk/services/CSIPhylogeny/, accessed on 1 November 2022)) that identifies single-nucleotide polymorphisms (SNPs) while also filtering and validating SNP positions before generating a phylogeny based on the concatenated alignment of high-quality SNPs. The SNP selections were conducted using the default parameters, including a minimum distance between SNPs (prune) as 10 bp, a 10× minimum depth at SNP positions, a minimum mapping quality of 25, a minimum Z-score of 1.96, and a minimum SNP quality of 30.

## 3. Results

### 3.1. Prevalence of Colistin-Resistant Isolates from Four Chinese Provinces

A total of 884 (41.6%) antibiotic-resistant isolates were identified from the 2126 collected isolates, and these exhibited a MIC range of 4–1024 mg/L (Table 1). The prevalence of resistant isolates from the Shandong Province was the highest (82.7%, 268/324), while the prevalence of resistant isolates from the Guangdong Province was the lowest (26.8%, 271/1011). Isolates from the Fujian and Jiangsu Provinces exhibited moderate prevalence rates. The resistance prevalence among sample types provided little evidence for region-specific patterns. Further analysis of MIC levels indicated that the resistance levels of isolates were relatively evenly distributed, with significant proportions of strains at all resistance levels (Figure 1).

16S rRNA gene sequencing revealed that 60.1% (205/341) of colistin-resistant bacteria were *Bacillus* spp., while *Enterobacter*, *Staphylococcus*, and *Aeromonas* spp. respectively accounted for 5.9% (20/341) of the isolates, and 22.3% (76/341) belonged to other genera (e.g., *Lactococcus*, *Proteus*, and *Vagococcus*; Appendix A).

### 3.2. Determination of MIC Value of Electron Transformants

Plasmids from strains including 11 of *Bacillus* sp. and one each of *Staphylococcus*, *Aeromonas*, and *Enterobacter* were successfully transformed into *E. coli* DH5α cells by electroporation. The transformants exhibited resistance to colistin at an MIC value of 4 mg/L, representing a four-fold increase over that of *E. coli* DH5α.

### 3.3. Antimicrobial Susceptibility of Bacillus *spp.*

Most recovered isolates were *Bacillus* spp., and thus, antimicrobial susceptibility analyses were conducted for these representative commensal bacteria (Figure 2). The resistance of *Bacillus* isolates to other antibiotics is shown in Figure 2. *Bacillus* isolates resistant to sulfonamides, quinolones, tetracyclines, β-lactams, and macrolides were observed. Most of the isolates were resistant to cefepime and ceftriaxone, with resistance rates of 53.2% and 52.7%, respectively. These strains also exhibited relatively high resistance towards ceftiofur (43.3%), trimethoprim-sulfamethoxazole (39.9%), meropenem (36%), ampicillin (26.6%), and tetracycline (26.1%). Few isolates were resistant to imipenem (4.4%), gentamicin (3%), and ciprofloxacin (1.5%).

An antibiogram of all antibiotics evaluated for the *Bacillus* isolates is shown in Appendix A, suggesting that approximately 69.3% of *Bacillus* spp. were MDR. MDR index under 202 *Bacillus* isolates were 0.08–1. The across region MDR index values were 0.08–0.75 in Guangdong Province, 0.08–0.67 in Fujian Province, 0.08–0.5 in Jiangsu Province, and 0.17–1 in Shandong Province, respectively. MDR index values ≥ 0.2 indicate a high-risk of an antibiotic-exposed source. 89% of the strains from the Fujian Province exhibited MDR index values > 0.2, which was the highest among the four provinces. Strains with MDR index values > 0.2 from the Guangdong and Jiangsu Provinces accounted for 63% and 58% of province totals, respectively. In addition, 44% of the Shandong Province isolates exhibited MDR index values > 0.2 (Figure 3).

Many isolates were resistant to β-lactam antibiotics (*n* = 86), followed by those resistant to β-lactams, fosfomycin, and tetracycline (*n* = 23) (Figure 4). Figure 5 shows the dispersion of antibiotic resistance for all isolated *Bacillus* strains. Except for Jiangsu province, the drug resistance dispersion of isolates from other regions was high.

### 3.4. Bacillus licheniformis ST Diversity

Considering that *B. licheniformis*, a widespread commensal bacterial, accounted for the largest proportion of drug-resistant *Bacillus* spp. in this study (28.3%, 58/205), this species was selected to determine the molecular diversity, and minimum spanning trees were generated based on MLST data (Figure 6). A total of six STs were identified among the 58 strains, with three STs being novel. Novel STs were submitted to the PubMLST database and were assigned the identifiers ST51, ST52, and ST53. Among the 58 isolates, the ST4 (*n* = 28) group accounted for the highest proportion from the Guangdong Province. The ST1 (*n* = 16) group was identified in all four provinces, while ST3 (*n* = 3) was identified in strains from the Jiangsu and Guangdong Provinces. Two newly identified ST51 strains were recovered from the Fujian Province. The novel ST52 and ST53 types were recovered from the Guangdong Province.

Some strains belonged to the same STs but had different drug resistance patterns (Figure 7). ST1 isolates from different provinces exhibited eight drug resistance profiles and were derived from shrimp, pond water, and aquafeed. Further, the drug resistance profiles of the three ST3 isolates from pond water and aquafeed were highly different. Moreover, 16 different antibiotic resistance spectra were identified among ST4 strains that were isolated from shrimp, pond waters, and soils.

A total of 235 *B. licheniformis* genomes were recovered from the GenBank database, including the ST1 (*n* = 16), ST3 (*n* = 56), and ST4 (*n* = 1) types (Appendix A). ST1 genomes derived from isolates from Japan, Denmark, Germany, Turkey, and China samples from honey, soils, coarse salts, and pig faeces. The ST3 genomes were derived from isolates recovered from mushroom soup, chicken, leaves, soil, heroin, faeces, Maotai Daqu, soybeans, cheese, seawater, goats, and air samples that were collected from Europe, Africa, and Asia. ST4 isolates were recovered from unknown sources from the Netherlands. Analysis of isolates from this study and those from previous studies indicated that ST1 and ST3 strains are frequently found in foods. These two ST types have been identified in many countries, indicating a wide distribution. Only one record of the ST4 type was identified in the database. Thus, new isolates recovered from the Guangdong province in this study can provide new information regarding ST4 *B. licheniformis* types. Further, these results confirm that *B. licheniformis* is widely distributed globally and that novel ST types remain to be discovered.

### 3.5. Antibiotic Resistance Genes of Bacillus licheniformis

The results of 17 *Bacillus licheniformis* in CARD are shown in Figure 8, Appendix A. The genes *BcIII* (resistant to cephalosporins) and *qacJ* (mediates resistance to quaternary ammonium compounds) were found in all strains, and the *ErmD* gene (macrolide-lincosamide-streptogramin B resistance element) was present in 16 isolates. Bacitracin resistance genes *bcrA*, *bcrB,* and *bcrC* are only in a few isolates from Fujian province (*n* = 2, 11.8%) and Guangdong province (*n* = 3, 17.6%). Genes encoding resistance to colistin and other antibiotics were not detected, which means the antibiotic phenotypes of the isolates were not really consistent with the antibiotic resistance genotypes.

### 3.6. Pathogenicity, Virulomes, and Mobile Genetic Elements

PathogenFinder analysis predicted that 17 of the *B. licheniformis* isolates are human pathogens, with probabilities ranging from 0.79 to 0.81 (Appendix A). Thus, *B. licheniformis* from shrimp farms can be potential pathogens in humans. A total of 22 virulence genes were identified in the genomes of 17 *B. licheniformis* isolates (Figure 9, Appendix A). The most frequently observed virulence determinants were *dhbF*, *clpP*, and *lap* that were present in 65%, 59%, and 59% of the isolates, respectively. Notably, 16 of the 17 isolates harboured at least 4–6 virulence determinants.

Diverse IS types belonging to different IS families were identified among isolates. The most frequently detected IS families were IS1182, IS3, IS1585, ISL3, IS110, and ISAs1 (Appendix A). The Tn3 transposon was only identified in two isolates from Guangdong Province soils.

### 3.7. Phylogenetic Lineages of Bacillus licheniformis

To evaluate genetic relationships among *B. licheniformis*, their core genome phylogenetic lineages from different sources and regions were evaluated (Figure 10). The analyses revealed that the strains were primarily divided into two lineage groups, with grouping by collection source or region not obvious, because both groups comprised isolates from different regions and sources. A close phylogenetic relationship was observed between isolates from pond water samples collected from the Fujian Province and aquafeed samples from the Jiangsu Province (FJ-ts50 & JS-YHS57; FJ-ts54 & JS-YHS52).

A phylogenetic tree was constructed including 73 genomes from the Genbank database and those from this study to explore phylogenetic associations of global *B. licheniformis*. Specifically, an SNP core-genome phylogenetic tree was constructed and analysed in the context of strain resistomes and virulomes (Figure 11). Three primary clades were present in the tree, all including subclades. The genomes produced in this study and from the NCBI database were distributed across the phylogeny and belonged to similar groups, with little evidence of clustering by region or source.

Two *B. licheniformis* isolates from shrimp (ST53) and pig faeces (ST1) that originated from China (GD-GWC10X-1) and Germany (GCF_007831435), respectively, exhibited high genetic similarity. Their resistomes were highly similar, while their virulomes differed. Isolates from other countries were also closely phylogenetically related. For example, GCF_007831655 with GCF_007832135 and GCF_006494795 with GCF_007831035 were observed in monophyletic subtrees.

## 4. Discussion

The eastern and southern coastal areas of China investigated in this study are major shrimp farming areas. Samples analysed here for colistin-resistant bacterial presence included shrimp and their growth environments. Thus, the investigation comprehensively analysed *Penaeus vannamei* aquaculture environments of China. Colistin resistance was identified in 41.6% of isolates retrieved from samples. The prevalence of colistin-resistant isolates slightly differed among provinces analysed in this study. Previous studies have observed colistin resistance rates of 65.2% and 16.5% in fisheries and waterfowl aquaculture environments from the Guangdong Province, respectively [12,14]. Further, an investigation of retail products in Beijing revealed a colistin resistance rate of bacterial isolates of 8.4% [31]. A similar analysis of colistin resistance in chicken-derived isolates of central China revealed a prevalence of 11.6% [32]. Thus, the prevalence of colistin resistance is highly related to the host species and region, suggesting a need to scale-up studies to better assess colistin resistance among commensal bacteria. Moreover, the farms analysed in this study did not use colistin in their practices, suggesting that other risk factors, such as using feed rich in colistin-resistant bacteria or even environmental factors, may play important roles in spreading resistance among aquatic environments.

16S rRNA gene sequencing indicated that *Bacillus* strains were abundant among the colistin-resistant commensal bacteria isolated from shrimp cultures. Iurlina, Saiz, Fuselli, Fritz and Technology [33] also found a high incidence of *Bacillus* spp. in the food samples examined. This may be due to the widespread use of *Bacillus* spp. as an aquaculture probiotic. *Bacillus* are commensal bacteria and natural members of the gut microbiota of some aquatic organisms. Anokyewaa, Amoah, Li, Lu, Kuebutornye, Asiedu and Seidu [15] reported that *Bacillus* spp. can enhance the immunity of aquatic organisms and improve their growth performance. Previous studies on colistin-resistant bacteria have mostly focused on opportunistic pathogens [9,11,12]. However, the results of this study suggest that focusing only on Gram-negative bacteria may not be comprehensive enough to inform control of antibiotic resistance. Rather, a more comprehensive understanding of colistin resistance should consider commensal bacteria.

The majority of bacteria are capable of incorporating foreign DNA into their genome, either integrating it or maintaining it in an episomal form. Multiple natural processes could result in this. As an illustration, numerous bacterial species exhibit the capacity to import foreign DNA from nearby lysed cells. Another typical method of gene transfer in bacteria is phage transduction. Electroporation is a common laboratory technique used to convert eukaryotic and prokaryotic cells using electrical pulses of high amplitude and brief duration in the presence of foreign DNA [34]. This transformation for bacteria is significantly simple, achieves a high transformation rate, and is widely applicable; consequently, it is often being used to introduce exogenous plasmids into recipient bacteria to transfer the corresponding genetic traits of donor cells. The high transferability of colistin resistance by electroporation transformation shown in this study further raises concerns about the spread of colistin resistance through commensal bacteria. Thus, when bacteria die and lyse, colistin resistance markers may transfer to other bacteria under certain environmental conditions. 16S rRNA gene sequencing revealed that Gram-negative and Gram-positive bacteria were among the colistin-resistant commensal bacteria isolated in this study. Furthermore, resistance in these bacteria was transferable through electroporation transformation. This implies that the widespread presence of commensal bacteria in environments may lead to the spread of drug resistance and considerable expansion of the resistance reservoir pool. Thus, commensal bacteria may be hotspots for the development of colistin resistance and spread in environments, and even to humans, thereby posing a threat to public health.

Multidrug susceptibility testing of *Bacillus* spp. revealed that approximately 69.3% of the isolates were multidrug-resistant. Antibiotics including sulfonamides, quinolones, tetracyclines, β-lactams, and macrolides have been widely used in Chinese aquaculture, consistent with the results of this study [35]. Overall, *Bacillus* isolates were most resistant to cephalosporin, with prevalence rates of >40%. Ceftriaxone-resistant commensal bacterial have been detected in humans, food animals, or environments [36]. Although cefepime and ceftriaxone are only intended for human use, ceftiofur, which is structurally similar to ceftriaxone, has been widely used in animal production [37]. Notably, the prevalence of resistance to the fourth-generation cephalosporin cefepime was significantly higher than that to the third-generation cephalosporins, including ceftiofur and ceftriaxone. Higher prevalence of resistance to newer generations of antibiotics suggests the ongoing need for continued surveillance and monitoring programs for antimicrobial use in the aquaculture sector and antimicrobial resistance of aquatic food isolates, in addition to the development of new medicines that can mitigate the drug resistance of existing antibiotics [20].

Water environments represent concerning reservoirs of antibiotic resistance and are home to many MDR bacteria. For example, Huang, Zhang, Tiu and Wang [5] demonstrated that most isolates recovered from fish and aquaculture environments were resistant to multiple antibiotics. Further, an additional study demonstrated that the use of antibiotics significantly and rapidly increased resistance levels, due to the elimination of susceptible strains and the overgrowth of resistant strains [38]. Further, MDR has been reported in the bacterial groups *Enterobacteriaceae*, *Pseudomonads*, *Vibrionaceae*, and *Klebsiella* within aquaculture settings [7]. Thus, high proportions of MDR bacteria require vigilant surveillance, because diseases caused by MDR bacteria can no longer be treated [39]. When MDR bacteria interact with other bacteria in environments, resistance determinants can be transferred to clinically important bacteria through horizontal gene transfer (HGT), representing a significant threat to human health. Thus, abundant MDR bacteria in aquatic environments must be carefully evaluated, and corresponding control measures should be taken [40,41]. The rigorous evaluation of the potential presence of antibiotic-resistant isolates from aquatic shrimp can avoid the possibility of horizontal transmission to other bacteria in the same food or to bacterial pathogens in the intestinal system [42].

Previous studies have shown that identical STs exhibit different resistance to different antibiotics, consistent with the results of this study [12,43]. Thus, antibiotic resistance among the same bacterial species showed a high degree of diversity, suggesting that they may have significant differences in antibiotic exposure genetic background [42]. Resistance-related genes associated with colistin and multidrug susceptibility test results were not identified by comparison against the CARD database, possibly due to an incomplete understanding of antibiotic resistance mechanisms, consistent with the results of Li, Yang, Hu, Wang, Rong, Li, Sun, Wang, Zhang and Wang [44]. The latter speculated that this discrepancy likely occurred because antibiotic-resistant bacteria harbour other less common ARGs that were not investigated or that there is an incomplete understanding of antibiotic resistance mechanisms. Many studies have observed such discrepancies between antimicrobial resistance phenotypes and antimicrobial resistance genes. Yu, Tang, Wang, Liao, Wang, Rong, Li, Ge, Gao, Dong and Safety [45] concluded that this may be due to the fact that the antibiotic phenotype can be expressed under the stimulus of many different genetic determinants. Alternatively, the antibacterial phenotype may be mediated not only by resistance genes, but also by membrane structure and physiological metabolism. In addition, the CARD database merely focuses on providing high-quality reference data and molecular sequences within a controlled vocabulary. Therefore, CARD database’s limitations should be acknowledged, although it is constantly being updated. It cannot detect all resistance genes, but is based only on the result of genetic sequences submitted by various researchers, existing ARG databases, and published papers. In summary, antibiotic resistance mechanisms are still rapidly evolving, and thus, additional research is needed to understand them.

*Bacillus licheniformis* is a widely distributed commensal bacterial species that is often found in food [46]. It is usually cited as conferring many benefits to hosts. However, several studies have demonstrated the negative effects of *B. licheniformis* on hosts, including food poisoning, disease, and virulence [46,47,48]. Some studies have even suggested that *B. licheniformis* may become a pathogen that causes human infection [49,50]. Both *qacJ* and *BcIII* antibiotic resistance genes were identified in *B. licheniformis* genomes retrieved from the NCBI database and identified in genomes produced in this study. Thus, these two genes could be typical antibiotic resistance genes of *B. licheniformis*. *ErmD* and *bcr* genes were also identified in some publicly available *B. licheniformis*, indicating that these antibiotic resistance genes may have been acquired either by mutation of existing genes or by heterologous acquisition of resistance genes from an external source. The identification of virulence factors is the key to evaluating bacterial pathogenicity. Its secreted microbial products have the ability to enter host cells and use them to aid in infection mechanisms [51].Virulence factors from *B. licheniformis* genomes were enriched in functions related to adherence, metal uptake, and secretion systems that may promote the long-term colonisation of hosts. The propagation and diversity of ARGs and VFs coexisting with pathogenicity among *B. licheniformis* strains likely suggest the potential for public health risks owing to their MDR and virulence traits.

Only a limited number of studies have been conducted on the association and drug resistance of globally prevalent *B. licheniformis*. The MLST analyses of this study suggest that drug-resistant *B. licheniformis* are widespread in animals, feed, and aquaculture farm environments. Isolates recovered from the Fujian Province exhibited close genetic relationships with those from the Jiangsu Province and may be more likely to spread through dispersal across close geographic areas. Waters, soils, and the aquatic product trade may play important roles in the spread of drug-resistant bacteria among different locations. wgSNP analysis identified similar genetic subtypes and lineage clusters as MLST analyses, indicating the high discriminatory power of the former [52]. Molecular typing of *B. licheniformis* from this study and those in the NCBI database indicated shared populations of isolates among different hosts, revealing high genetic diversity and close genetic relationships among isolates. Further, a distinct phylogenetic cluster including strains from the Guangdong Province of China and from Germany suggested the possibility of long-distance transmission. Strains carrying the resistance genes *BcIII* and *qacJ* are generalists and can colonise different species. Thus, *B. licheniformis* may be dispersing across relatively large global distances. Consequently, frequent international food or animal trades, in addition to colonised or infected humans, might facilitate the distribution of commensal bacteria. The global ubiquity of *B. licheniformis* combined with the pathogenicity, virulence factor, and mobile genetic element distributions suggests a high potential for public health risks from commensal bacteria that should not be ignored.

## 5. Conclusions

Here, we demonstrate that commensal bacteria from aquatic shrimp exhibit high antibiotic resistance prevalence and that colistin resistance may be transferable under certain conditions. Thus, investigations of colistin resistance among bacteria should not only focus on opportunistic pathogens but also on commensal bacteria. These data provide further insights into the antibiotic resistance of commensal bacteria in Chinese aquatic shrimp. In addition, the high levels of MDR identified in this study suggest the need for enhanced surveillance and policies to limit the use of antibiotics to avoid further spread of antibiotic resistance. As shrimp is a widespread aquatic food, there is a need to explore and re-evaluate its role as a possible reservoir of transmissible antibiotic resistance. The State Aquatic Product General Bureau should consider and implement antibiotic susceptibility screening to ensure aquatic food safety and limit the spread of antibiotic-resistant strains. Moreover, an incomplete understanding of resistance mechanisms suggests that additional research on antibiotic resistance mechanisms is needed. In addition, the identification of genetically similar *B. licheniformis* isolates from distant regions and diverse sources demonstrates the possibility of their widespread persistence. Some commensal bacteria can be harmful to hosts, although they are widely distributed and have long been considered beneficial. Thus, greater research is needed to more comprehensively understand commensal bacteria and their antibiotic resistance mechanisms to inform countermeasures.

## Figures and Tables

**Figure 1 foods-12-02143-f001:**
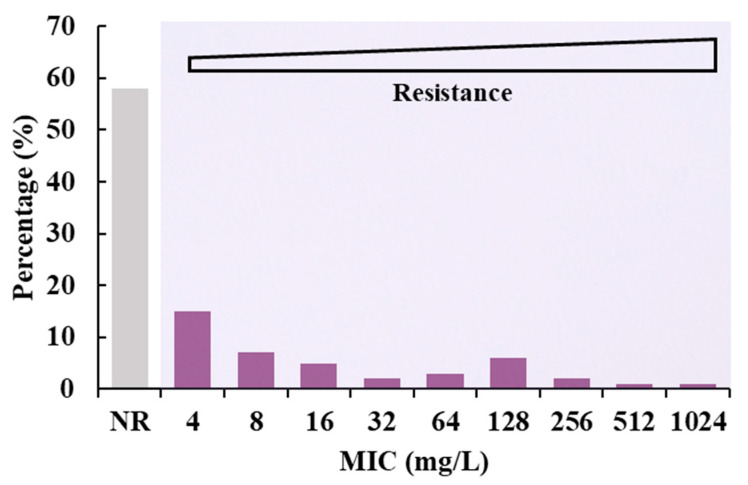
Levels of resistance to colistin antibiotics among recovered isolates. NR, non-resistant.

**Figure 2 foods-12-02143-f002:**
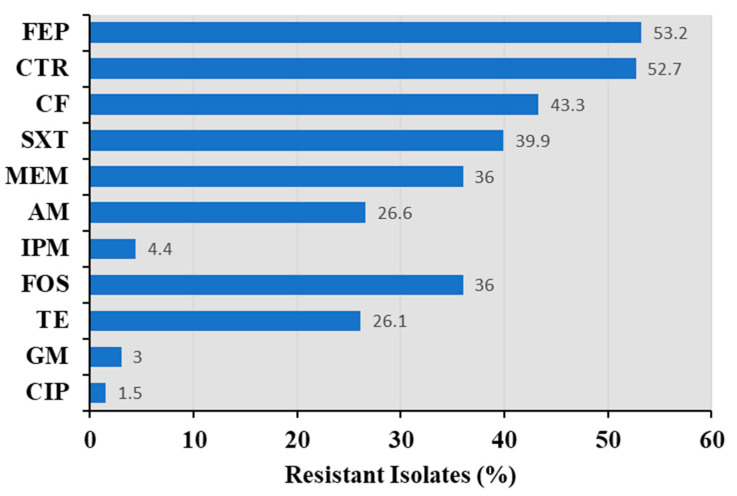
Resistance of *Bacillus* isolates to different antibiotics. β-lactams: FEP, cefepime; CTR, ceftriaxone; CF, ceftiofur; SXT, trimethoprim-sulfamethoxazole; MEM, meropenem; AM, ampicillin; and IPM, imipenem. FOS, fosfomycin. TE, tetracycline. GM, gentamicin (aminoglycoside). CIP, ciprofloxacin (quinolone).

**Figure 3 foods-12-02143-f003:**
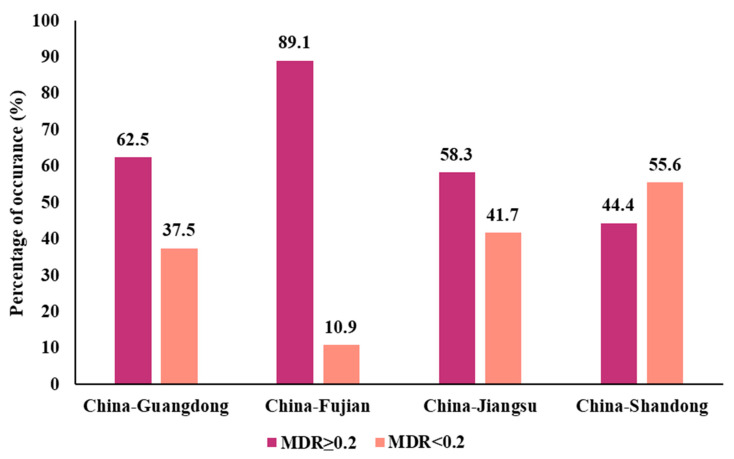
Prevalence of *Bacillus* strains with MDR index values ≥ 0.2 or MDR index values < 0.2.

**Figure 4 foods-12-02143-f004:**
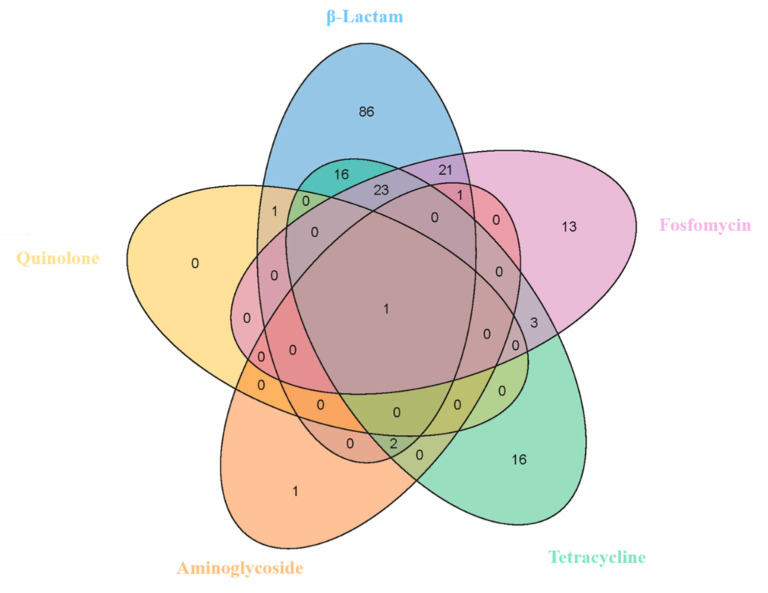
Venn diagram showing the multidrug resistance characteristics of *Bacillus* spp. β-Lactams: FEP, cefepime; CF, ceftiofur; SXT, trimethoprim-sulfamethoxazole; MEM, meropenem; CTR, ceftriaxone; AM, ampicillin; IPM, imipenem. FOS, fosfomycin. TE, tetracycline. GM, gentamicin (aminoglycoside). CIP, ciprofloxacin (quinolone).

**Figure 5 foods-12-02143-f005:**
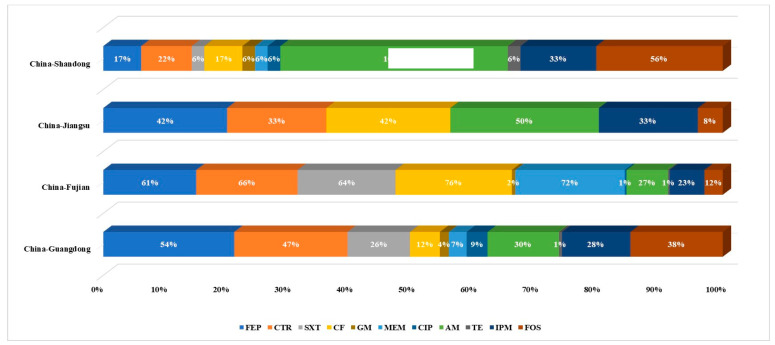
The dispersion of antibiotic resistance in each region. FEP, cefepime; CTR, ceftriaxone; SXT, trimethoprim-sulfamethoxazole; CF, ceftiofur; GM, gentamicin; MEM, meropenem; CIP, ciprofloxacin; AM, ampicillin; TE, tetracycline; IPM, imipenem; FOS, fosfomycin.

**Figure 6 foods-12-02143-f006:**
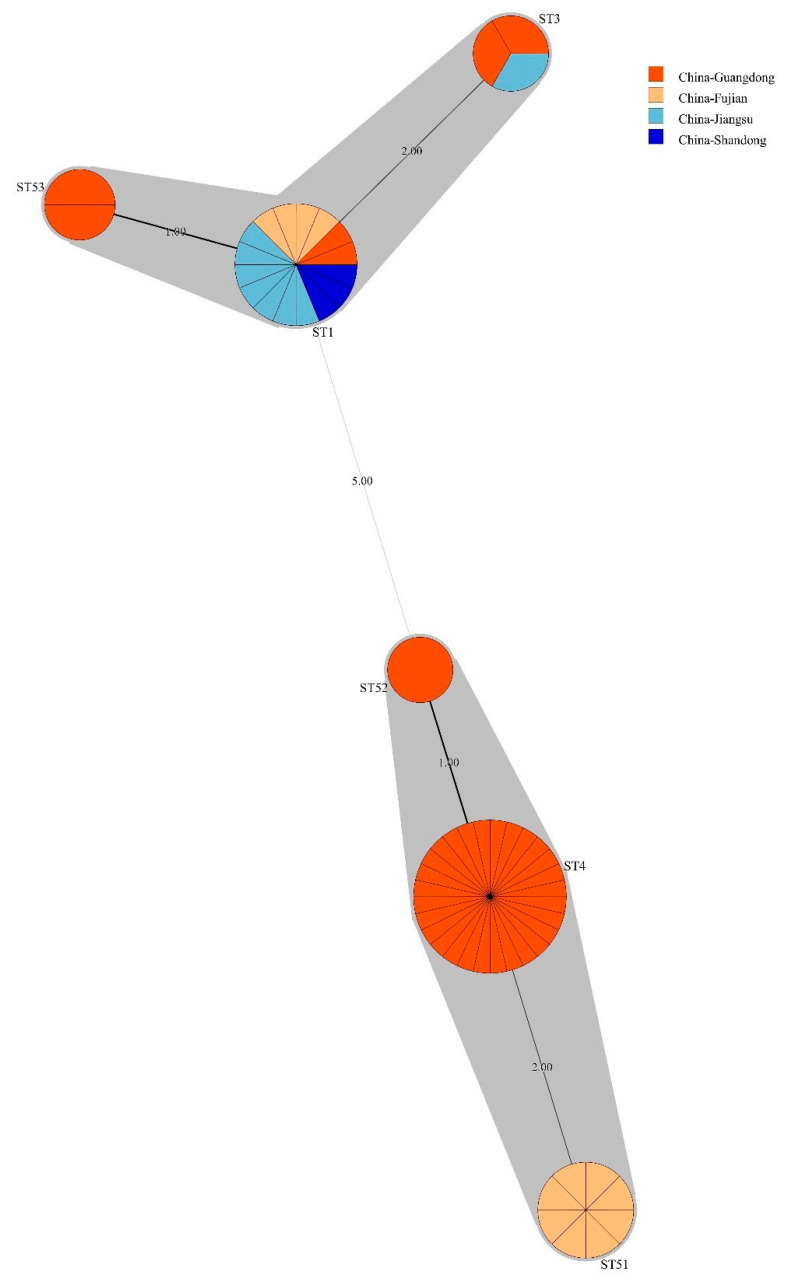
Minimum spanning trees for *Bacillus licheniformis* based on multilocus sequence data. Each node within the tree represents a single sequence type (ST), and the node size is proportional to the number of isolates represented by the node. Different node colours indicate different isolate origins. The length of branches between nodes is proportional to the number of different alleles (among six MLST genes) that differ between two linked nodes/STs.

**Figure 7 foods-12-02143-f007:**
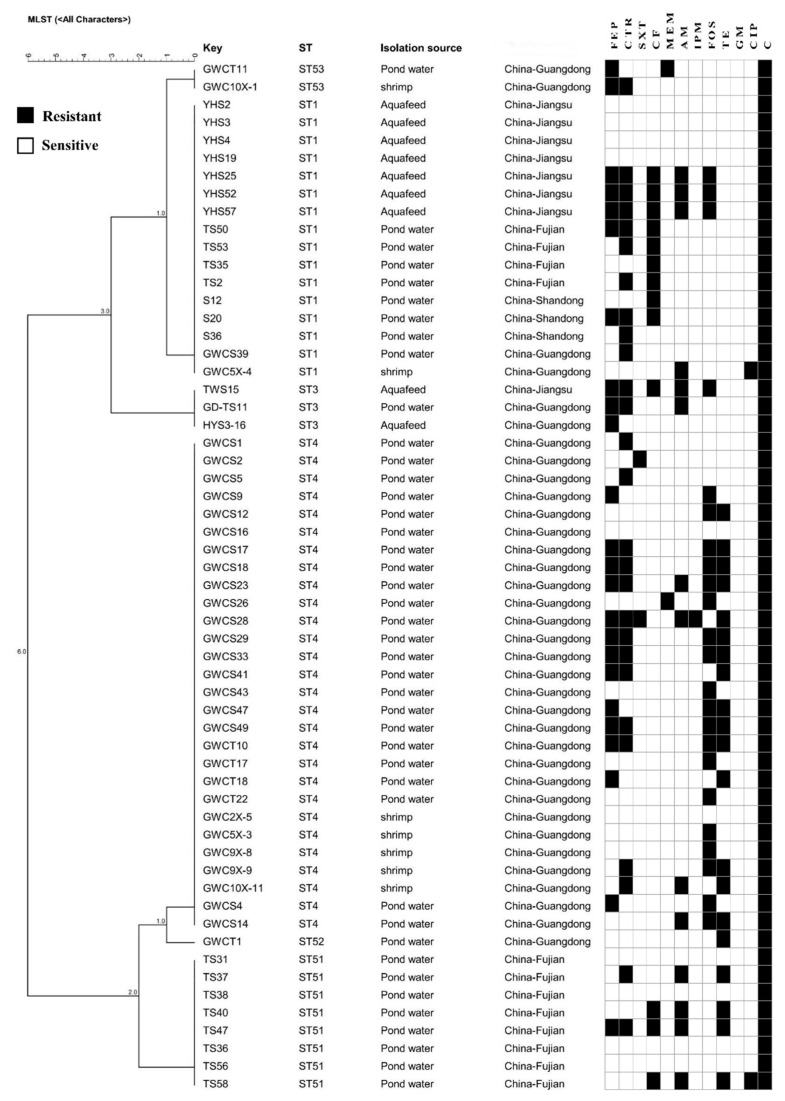
MLST sequence-based phylogenetic tree of *Bacillus licheniformis* strains. Phylogeny of 58 *B. licheniformis* isolates and their source, region of isolation, and resistome characteristics. ST, sequence type; FEP, cefepime; CTR, ceftriaxone; SXT, trimethoprim-sulfamethoxazole; CF, ceftiofur; MEM, meropenem; AM, ampicillin; IPM, imipenem; FOS, fosfomycin; TE, tetracycline; GM, gentamicin; CIP, ciprofloxacin; and C, colistin.

**Figure 8 foods-12-02143-f008:**
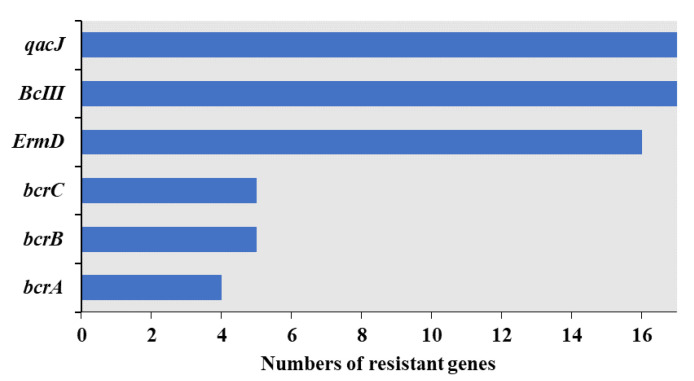
Antibiotic resistance gene abundances in *Bacillus licheniformis* genomes.

**Figure 9 foods-12-02143-f009:**
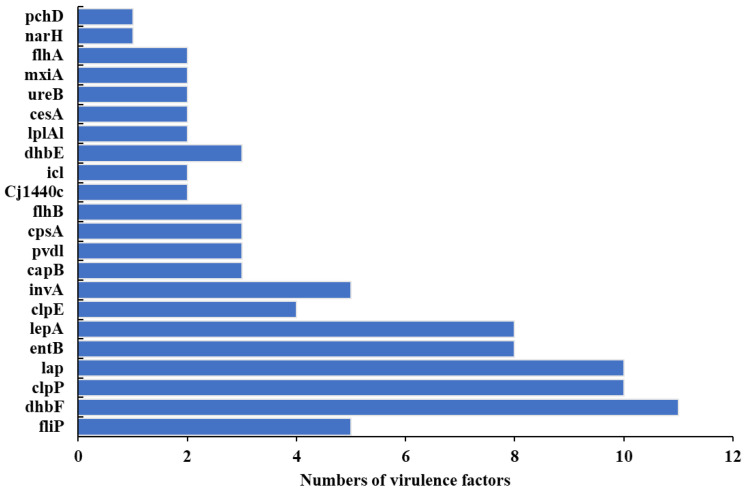
Virulence factor abundances in *Bacillus licheniformis*.

**Figure 10 foods-12-02143-f010:**
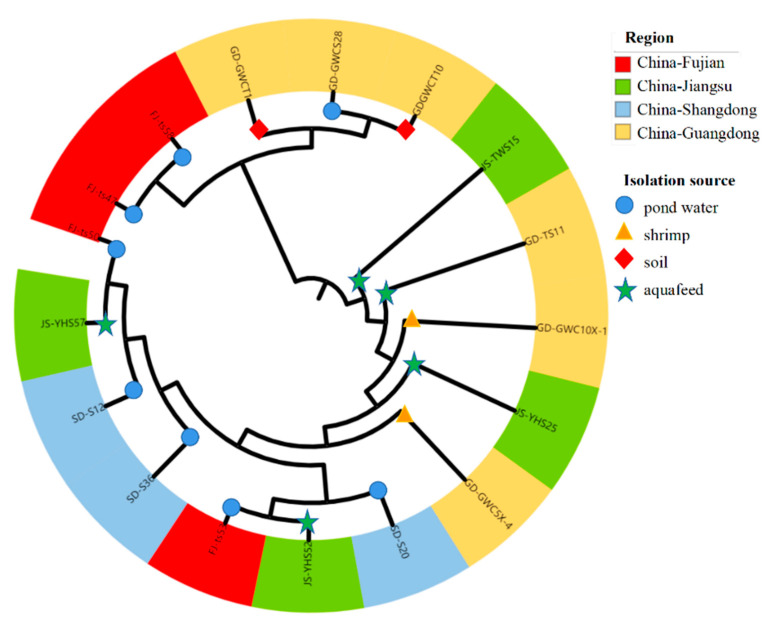
Phylogenetic tree of 17 *Bacillus licheniformis* isolates. Source information is identified by symbols at the end of branches. Coloured bars on the outer layer indicate isolation locations.

**Figure 11 foods-12-02143-f011:**
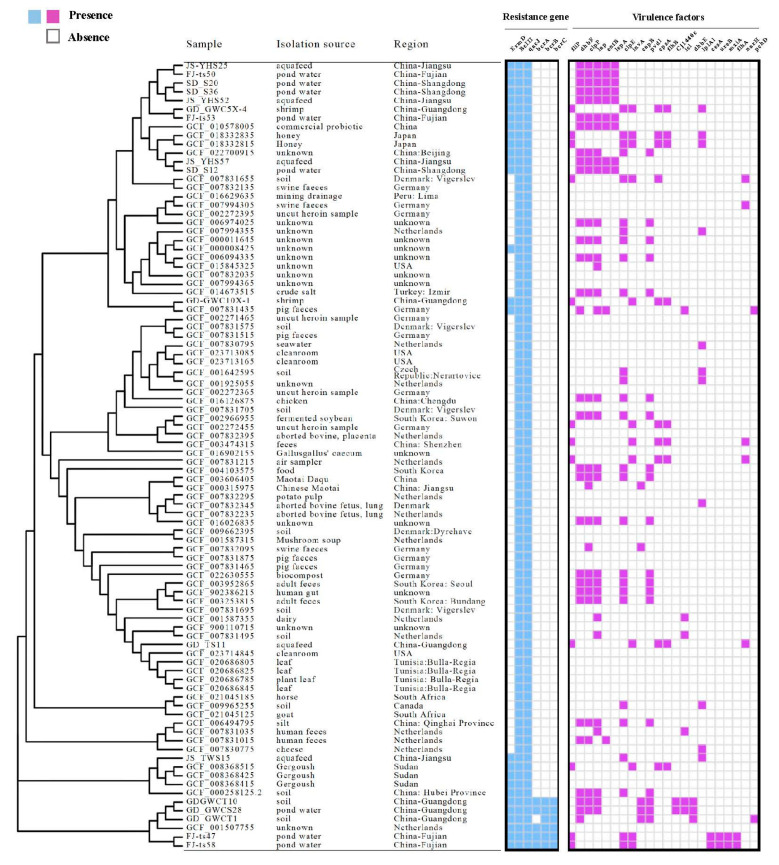
Phylogenetic tree of *Bacillus licheniformis* isolates. The tree was constructed using core-genome single nucleotide polymorphism (SNP) data and is mid-point rooted. The molecular characteristics of each isolate, including resistance genes and virulence-associated genes, are indicated by filled (presence) and empty (absence) squares.

**Table 1 foods-12-02143-t001:** Prevalence of colistin-resistant isolates in aquaculture samples.

Sample Site	Sample Type	Number (%)		Regional Drug Resistance Rate (%)
		Isolates Investigated	Resistant Isolates	
China— Guangdong	*Penaeus vannamei*	385	73 (19.0)	26.8
	Soil	183	62 (33.9)	
	Aquafeed	169	45 (26.6)	
	Pond water	274	91 (33.2)	
China— Fujian	*Penaeus vannamei*	222	88 (39.6)	41.0
	Soil	98	26 (26.5)	
	Aquafeed	115	62 (53.9)	
	Pond water	106	46 (43.4)	
China— Jiangsu	*Penaeus vannamei*	92	22 (23.9)	49.2
	Soil	58	31 (53.4)	
	Aquafeed	76	66 (86.8)	
	Pond water	24	4 (16.7)	
China— Shandong	*Penaeus vannamei*	120	115 (95.8)	82.7
	Soil	60	43 (71.7)	
	Aquafeed	84	60 (71.4)	
	Pond water	60	50 (83.3)	
Total		2126	884 (41.6)	

## Data Availability

The data presented in this study are available on request from the corresponding author.

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
