# Peer review of "Survey of Colistin Resistance in Commensal Bacteria from Penaeus vannamei Farms in China"

_foods, 2023, doi:10.3390/foods12112143_

Round 1

Reviewer 1 Report

The manuscript is interesting and has a certain significance regarding the public health by addressing the problem of transferable antibiotic resistances among the commensals in the food chain. The results show that shrimp farms can be a reservoir of antibiotic resistant bacterial strains, and that particularly Bacillus species are associated with colistin resistance. 

There are, however, certain points that the authors should address:

1) The species distribution among the colistin resistant Bacilli. Bacillis licheniformis was chosen for a closer inspection. The specific reason for that should be elucidated

2) Table 1 should be more clear. For example, counting from  the number of isolates from Guandond was 1213, of which 359 were colistin resistant. According to the text, line 204 these figures were 1011 and 271. Please, check the whole table. 

3) Were the presence of the plasmids used in the electroporation experiments (Sections 2.4 and 3.2) physically demonstrated in the recipients. Were any of B. licheniformis strains subjected to MLST included among the donors? 

4) Lines 398 - 411: While electroporation is an efficient laboratory method for horizontal gene transfer. The authors should discuss bacterial conjugation and transformation as the more relevant ways of antibiotic resistances to spread in nature. Were the antuibiotic resistance plasmids used in the experiments analysed for genes associated with conjugation or mobilization. 

The English language and terminology are generally understandable. However, a linguistic revision preferably by a native speaker is recommended. 

Author Response

Reviewer 1:

  1. Response to comment:

The species distribution among the colistin resistant Bacilli. Bacillis licheniformis was chosen for a closer inspection. The specific reason for that should be elucidated.

Response:

Thank you for your suggestions.

We agree with your comment and have revised the sentence according to your comment as follows.

Considering that B. licheniformis, a widespread commensal bacterial, accounted for the largest proportion of drug-resistant Bacillus spp. in this study (28.3%, 58/205), this species was selected for determine the molecular diversity and minimum spanning trees were generated based on MLST data (Fig. 6). (page 15, line 280-283)

  1. Response to comment:

Table 1 should be more clear. For example, counting from the number of isolates from Guandond was 1213, of which 359 were colistin resistant. According to the text, line 204 these figures were 1011 and 271. Please, check the whole table.  

Response:

Thank you for your suggestion.

After checking the full text carefully, we did not find the mistake of “the number of isolates from Guandond was 1213, of which 359 were colistin resistant” as you said. According to your suggestion, we have revised Table 1 to make the data display clearer. (page 7-8)

  1. Response to comment:

Were the presence of the plasmids used in the electroporation experiments (Sections 2.4 and 3.2) physically demonstrated in the recipients. Were any of B. licheniformis strains subjected to MLST included among the donors?

Response:

Thank you for your suggestion.

We are very sorry for the unclear expression.

After the bacteria die and lyse, the DNA is exposed to the environment and can be taken up by the receptor. If ingested DNA carries resistant segments, it may transfer resistant fragments to the recipient, causing the recipient bacteria to acquire resistance. Electroporation is a highly effective method in laboratory for introducing exogenous nucleic acids into many cell types, including bacterial and mammalian cells.

We extracted the plasmids from colistin resistant bacteria and transferred the plasmids into Escherichia coli DH5α by electroshock transformation. We have confirmed that the minimum inhibitory concentration of the recipient bacterium Escherichia coli DH5α to colistin is lower than 2 mg/L, that is, it is not resistant. After electroporation, the minimum inhibitory concentration of the recipient bacteria to colistin was 4 mg/L, that is to say, they showed drug resistance. Therefore, we believe that during the electroporation, the drug-resistant bacteria, as the donor bacteria, transferred the colistin-resistant fragments on plasmids to the recipient bacteria.

We used 16S rRNA gene sequencing to reveal the genera of colistin resistant commensal bacteria and found that 60.1% (205/341) were Bacillus spp., while Enterobacter, Staphylococcus, and Aeromonas spp. respectively accounted for 5.9% (20/341) of the isolates, and 22.3% (76/341) belonged to other genera. Based on this result, 20 Bacillus strains, three Staphylococcus aureus strains, two Aeromonas strains, and three Escherichia coli strains were selected as representatives for the electroshock transformation. In the 20 Bacillus strains, there is one B. licheniformis strain subjected to MLST included among the donors.

  1. Response to comment:

Lines 398 - 411: While electroporation is an efficient laboratory method for horizontal gene transfer. The authors should discuss bacterial conjugation and transformation as the more relevant ways of antibiotic resistances to spread in nature. Were the antuibiotic resistance plasmids used in the experiments analysed for genes associated with conjugation or mobilization.

Response:

Thank you for your suggestion.

We agree with your suggestion and have revised the sentence according to your comment as follows.

The majority of bacteria are capable of incorporating foreign DNA into their genome, either integrating it or maintaining it in an episomal form. Multiple natural processes could result in this. As an illustration, numerous bacterial species exhibit the capacity to import foreign DNA from nearby lysed cells. Another typical method of gene transfer in bacteria is phage transduction. Electroporation is a common laboratory technique used to convert eukaryotic and prokaryotic cells using electrical pulses of high amplitude and brief duration in the presence of foreign DNA (Montes-Horcasitas et al., 2004). This transformation for bacteria is significantly simple, achieves a high transformation rate, and is widely applicable; consequently, it is often used to introduce exogenous plasmids into recipient bacteria to transfer the corresponding genetic traits of donor cells. (page 21-22, line 402-412)

       We presented conjugation- or mobilization-related virulence factors and mobile genetic elements in colistin resistant bacteria in Section 3.6. It showed that resistant bacteria carry not only virulence factors associated with adhesion and metastasis, but also some mobile genetic elements that can transfer horizontally across microbial communities.

Reviewer 2 Report

The authors present an interesting manuscript, on a very actual topic – antibiotic resistance within the One Health framework. It is generally well built, but English needs improving throughout the whole text, not only in the examples given below.

Detailed suggestions follow.

Line 23 – no need for hyphen in colistin-resistance

Line 68 – enterococci should not be italicized, as it is not the taxonomic name of the genus (Enterococcus), but rather the common name

Line 112 – parentheses are not necessary in standard 20776-1

Lines 127 – 128 – “We selected 20 Bacillus strains, three Staphylococcus aureus strains, two 127 Aeromonas strains, and three Escherichia coli strains via 16S rRNA gene sequencing” – this sentence is not clear. What do the authors mean with “we selected … via 16SrRNA sequencing”? Do they mean that selection was based on 16SrRNA sequencing? If so, how? Did the authors mean that they merely used 16SrRNA sequencing for identifying the strains?

Line 140 – Fosfomycin does not need a capital initial

Lines 145 – 147 – What do the authors mean by “the low dose of antibiotic usage”? And by “high dose of antibiotic used in the system”? Which system do they mean? Also, this sentence needs improving English.

Line 324 – Please rephrase in a clearer manner “with mean probabilities ranging from …”. Probabilities of what?

Lines 327 and 328 – Please italicize the gene designations; further down in text, there are again some italics missing.

Lines 442 – 445 – Please rephrase, correcting grammar and sentence structure.

Lines 456 – 463 – CARD will not detect all antibiotic genes, just those whose sequences are already deposited there, and even then, only if they are conserved enough between bacteria. The limitations of this database should be acknowledged as well.

Supplemental files: The word "bacterias" does not exist; the singular of this word is bacterium and the plural is bacteria.

English needs revision. A few sentences have grammar and/or structure issues that need to be fixed. Revision should be done throughout the entire text, not only in the examples given.

Author Response

Reviewer 2:

  1. Response to comment:

Line 23 – no need for hyphen in colistin-resistance.

Response:

Thank you for your suggestion.

We agree with your suggestion and have revised the sentence according to your comment as follows.

Here, several shrimp farms were investigated to identify colistin resistance among commensal bacteria of aquaculture. (page 1-2, line 24-25)

  1. Response to comment:

Line 68 – enterococci should not be italicized, as it is not the taxonomic name of the genus (Enterococcus), but rather the common name.

Response:

Thank you for your suggestion.

We are very sorry for the inaccurate use of the word. We have revised the part in the manuscript as follows.

Studies on colistin resistance have primarily focused on opportunistic pathogens like Enterobacteriaceae and Enterococci, while commensal bacteria are rarely evaluated (Kim et al., 2021; Lv et al., 2022; Pan et al., 2022). (page 3, line 68-70)

  1. Response to comment:

Line 112 – parentheses are not necessary in standard 20776-1.

Response:

Thank you for your suggestion.

We agree with your comment and have revised the sentence according to your comment as follows.

The broth microdilution (BMD) method was then used based on the ISO standard 20776-1 (Standardization, 2006). (page 5, line 112-113)

  1. Response to comment:

Lines 127 – 128 – “We selected 20 Bacillus strains, three Staphylococcus aureus strains, two 127 Aeromonas strains, and three Escherichia coli strains via 16S rRNA gene sequencing” – this sentence is not clear. What do the authors mean with “we selected … via 16SrRNA sequencing”? Do they mean that selection was based on 16SrRNA sequencing? If so, how? Did the authors mean that they merely used 16SrRNA sequencing for identifying the strains?

Response:

Thank you for your suggestion.

We are very sorry for the confusion caused by the unclear expression and have revised the sentence according to your comment as follows.

We selected 20 Bacillus strains, three Staphylococcus aureus strains, two Aeromonas strains, and three Escherichia coli strains from colistin resistant bacteria. (page 5, line 128-129)

We identified colistin resistant bacteria according to the minimum inhibitory concentrations and used 16S rRNA sequencing to distinguish the drug-resistant bacteria strains. After identifying the type of bacteria, we selected several different strains for electroporation transformation, including 20 Bacillus strains, three Staphylococcus aureus strains, two Aeromonas strains, and three Escherichia coli strains.

  1. Response to comment:

Line 140 – Fosfomycin does not need a capital initial

Response:

Thank you for your suggestion.

We are very sorry for the inaccurate use of the word. We have revised the part in the manuscript as follows.

The BMD method was used to assess fosfomycin resistance. (page 6, line 141)

  1. Response to comment:

Lines 145 – 147 – What do the authors mean by “the low dose of antibiotic usage”? And by “high dose of antibiotic used in the system”? Which system do they mean? Also, this sentence needs improving English.

Response:

Thank you for your suggestion.

We are very sorry for the confusion caused by the unclear expression and have revised the sentence according to your comment as follows.

The multidrug resistance (MDR) index was calculated as the ratio of the number of antibiotics that the strains were resistant towards divided by the total number of evaluated antibiotics. The MDR value is typically equal to or less than 0.2 when antibiotics are used infrequently or at low doses to treat animals. Contrarily, using antibiotics more frequently or at a higher risk of exposure will result in a MDR index score that is greater than 0.2 (Hemamalini et al., 2022a). (page 6, line 144-149)

  1. Response to comment:

Line 324 – Please rephrase in a clearer manner “with mean probabilities ranging from …”. Probabilities of what?

Response:

Thank you for your suggestion.

We agree with your comment and have revised the sentence according to your comment as follows.

PathogenFinder analysis predicted that 17 of the B. licheniformis isolates are human pathogens, with probabilities of being a human pathogen ranging from 0.79 to 0.81 (Table S5). (page 17, line 327-328)

The data of "probability of being a human pathogen" is directly from the result with PathogenFinder. PathogenFinder is a web server that, by analyzing the user-uploaded bacterial whole genome sequence data, identifies genomic features associated with both pathogenicity and non-pathogenicity, so that it quickly predicts the bacteria’s potential pathogenicity.

It means that B. licheniformis, as a commensal bacterium, may pose a risk of disease to humans.

  1. Response to comment:

Lines 327 and 328 – Please italicize the gene designations; further down in text, there are again some italics missing.

Response:

Thank you for your suggestion.

We are very sorry for the inaccuracy of the gene designations. We have revised the part in the manuscript as follows.

The most frequently observed virulence determinants were dhbF, clpP, and lap that were present in 65%, 59%, and 59% of the isolates, respectively. (page 17, line 331-332)

  1. Response to comment:

Lines 442 – 445 – Please rephrase, correcting grammar and sentence structure.

Response:

Thank you for your suggestion.

We are very sorry for the inappropriate grammar and sentence structure. We have revised the part in the manuscript as follows.

The rigorous evaluation of the potential presence of antibiotic-resistant isolates from aquatic shrimp can avoid the possibility of horizontal transmission to other bacteria in the same food or to bacterial pathogens in the intestinal system (Haryani et al., 2023). (page 23, line 454-457)

  1. Response to comment:

Lines 456 – 463 – CARD will not detect all antibiotic genes, just those whose sequences are already deposited there, and even then, only if they are conserved enough between bacteria. The limitations of this database should be acknowledged as well.

Response:

Thank you for your suggestion.

We agree with your comment and have revised the sentence according to your comment as follows.

What's more, CARD database merely focuses on providing high-quality reference data and molecular sequences within a controlled vocabulary. Therefore, CARD database's limitations should be acknowledged, although it is constantly being updated. It can not detect all resistance genes, but is based only on the result of genetic sequences submitted by various researchers, existing ARG databases, and published papers. (page 24, line 473-478)

  1. Response to comment:

Supplemental files: The word "bacterias" does not exist; the singular of this word is bacterium and the plural is bacteria.

Response:

Thank you for your suggestion.

We are very sorry for the misspelled word. We have revised the part in the supplemental file as follows.

Table S2. 16sRNA results of some drug-resistant bacteria (Supplemental file: page 4)